# Binding and transport of D-aspartate by the glutamate transporter homolog Glt$_{Tk}$

Valentina Arkhipova, Gianluca Trinco, Thijs W Ettema, Sonja Jensen, Dirk J Slotboom*, Albert Guskov*

Groningen Biomolecular Sciences and Biotechnology Institute, Zernike Institute for Advanced Materials, University of Groningen, Groningen, The Netherlands

**Abstract** Mammalian glutamate transporters are crucial players in neuronal communication as they perform neurotransmitter reuptake from the synaptic cleft. Besides L-glutamate and L-aspartate, they also recognize D-aspartate, which might participate in mammalian neurotransmission and/or neuromodulation. Much of the mechanistic insight in glutamate transport comes from studies of the archeal homologs Glt$_{Ph}$ from *Pyrococcus horikoshii* and Glt$_{Tk}$ from *Thermococcus kodakarensis*. Here, we show that Glt$_{Tk}$ transports D-aspartate with identical Na$^+$: substrate coupling stoichiometry as L-aspartate, and that the affinities ($K_d$ and $K_m$) for the two substrates are similar. We determined a crystal structure of Glt$_{Tk}$ with bound D-aspartate at 2.8 Å resolution. Comparison of the L- and D-aspartate bound Glt$_{Tk}$ structures revealed that D-aspartate is accommodated with only minor rearrangements in the structure of the binding site. The structure explains how the geometrically different molecules L- and D-aspartate are recognized and transported by the protein in the same way.
DOI: https://doi.org/10.7554/eLife.45286.001

*For correspondence:
d.j.slotboom@rug.nl (DJS);
a.guskov@rug.nl (AG)

**Competing interests:** The authors declare that no competing interests exist.

## Introduction

Mammalian excitatory amino acid transporters (EAATs) are responsible for clearing the neurotransmitter glutamate from the synaptic cleft (for review see *Grewer et al., 2014*; *Takahashi et al., 2015*; *Vandenberg and Ryan, 2013*). EAATs are secondary transporters that couple glutamate uptake to co-transport of three sodium ions and one proton and counter-transport of one potassium ion (*Levy et al., 1998*; *Owe et al., 2006*; *Zerangue and Kavanaugh, 1996*). EAATs transport L-glutamate, L- and D-aspartate with similar affinity (*Arriza et al., 1994*).

D-aspartate is considered as a putative mammalian neurotransmitter and/or neuromodulator (*Brown et al., 2007*; *D'Aniello et al., 2011*; *Spinelli et al., 2006*) (reviewed in *D'Aniello, 2007*; *Genchi, 2017*; *Ota et al., 2012*). Such a role is also proposed for L-aspartate (*Cavallero et al., 2009*), however this is still a matter of debate (*Herring et al., 2015*). Both stereoisomers bind to and activate N-methyl-D-aspartate receptors (NMDARs) (*Patneau and Mayer, 1990*) and might be involved in learning and memory processes (reviewed in *Errico et al., 2018*; *Errico and Usiello, 2017*; *Katane and Homma, 2011*; *Ota et al., 2012*).

Although it is well established that EAATs take up D-aspartate (*Arriza et al., 1994*; *Gundersen et al., 1993*), structural insight in the binding mode of the enantiomer is lacking. The best structurally characterized members of the glutamate transporter family are the archeal homologs Glt$_{Ph}$ and Glt$_{Tk}$ (*Akyuz et al., 2015*; *Boudker et al., 2007*; *Guskov et al., 2016*; *Jensen et al., 2013*; *Reyes et al., 2013*; *Reyes et al., 2009*; *Scopelliti et al., 2018*; *Verdon et al., 2014*; *Verdon and Boudker, 2012*; *Yernool et al., 2004*), which share 32–36% sequence identity with eukaryotic EAATs (*Jensen et al., 2013*; *Slotboom et al., 1999*; *Yernool et al., 2004*). In contrast to EAATs, Glt$_{Ph}$ and Glt$_{Tk}$ are highly selective for aspartate over glutamate, and couple uptake only to co-transport of three sodium ions (*Boudker et al., 2007*; *Groeneveld and Slotboom, 2010*;

Guskov et al., 2016). Despite these differences, the amino acid residues in the substrate-binding sites of mammalian and prokaryotic glutamate transporters are highly conserved (Boudker et al., 2007; Jensen et al., 2013). The first structures of human members of the glutamate transporter family (Canul-Tec et al., 2017; Garaeva et al., 2018), showed that the substrate-binding sites are indeed highly similar among homologs (Figure 2—figure supplement 1).

Here, we present the structure of Glt$_{Tk}$ with the enantiomeric substrate D-aspartate. The crystal structure was obtained in the outward-facing state with the substrate oriented in a very similar mode as L-aspartate, showing that the two enantiomers bind almost identically regardless of the mirrored spatial arrangement of functional groups around the chiral Cα atom.

## Results

### Affinity of D-aspartate and stoichiometry of sodium binding to Glt$_{Tk}$

Using Isothermal Titration Calorimetry (ITC), we determined the binding affinities of D-aspartate to Glt$_{Tk}$ in the presence of varying concentrations of sodium ions (Figure 1A, Table 1). The affinity of the transporter for D-aspartate was strongly dependent on the concentration of sodium, similar to what has been reported for L-aspartate binding to Glt$_{Ph}$ and Glt$_{Tk}$ (Boudker et al., 2007; Hänelt et al., 2015; Jensen et al., 2013; Reyes et al., 2013). At high sodium concentration (500 mM), the $K_d$ values of Glt$_{Tk}$ for D- and L-aspartate binding level off to 374 ± 30 nM and 62 ± 3 nM, respectively. The ΔH values for binding of both substrates were favorable, with a more negative value of ~1 kcal mol$^{-1}$ for L-aspartate, indicating a better binding geometry for L- than for D-aspartate. For both substrates, the ΔS contribution was unfavorable (Table 1). When plotting the observed $K_d$ values for L- and D-aspartate against the sodium concentration (on logarithmic scales), the slopes of both curves in the lower limit of the sodium concentration are close to −3, indicating that binding of both compounds is coupled to the binding of three sodium ions (Boudker et al., 2007; Lolkema and Slotboom, 2015) (Figure 1B).

To test whether D-aspartate is a transported substrate, purified Glt$_{Tk}$ was reconstituted into proteoliposomes and uptake of [$^3$H]-D-aspartate was assayed. Glt$_{Tk}$ catalyzed transport of the radiolabeled substrate into the proteoliposomes. The $K_m$ for transport was 1.1 ± 0.11 µM at a sodium concentration of 100 mM (Figure 1C). This value is comparable to the $K_m$ for L-aspartate uptake under the same conditions (0.75 ± 0.17 µM). The stoichiometry Na$^+$: D-aspartate was determined by flux measurements of radiolabeled D-aspartate at different membrane voltages (Fitzgerald et al., 2017). Depending on the concentrations of Na$^+$ and D-aspartate on either side of the membrane, the imposed voltages either lead to flux of radiolabeled D-aspartate across the membrane (accumulation into or depletion from the lumen), or does not cause net flux (when the voltage equals the equilibrium potential) (Fitzgerald et al., 2017). The equilibrium potentials for different possible stoichiometries are calculated by:

$$E_{rev} = -\frac{60 mV}{\frac{n}{m}-1}\left(\frac{n}{m}log\frac{[Na^+]_{in}}{[Na^+]_{out}} + log\frac{[S]_{in}}{[S]_{out}}\right)$$

where $n$ and $m$ are the stoichiometric coefficients for Na$^+$ and substrate S, respectively. Membrane voltages were chosen that would match the equilibrium potential for stoichiometries of 2:1 (−78 mV), 3:1 (−39 mV) or 4:1 (−26 mV), and flux of radiolabeled D-aspartate was measured (Figure 1D). At −78 mV D-aspartate was taken up into the lumen; at −26 mV it was released from the liposomes; and at −39 mV there was little flux. From these data, we conclude that D-aspartate is most likely symported with three sodium ions. However, the flux was not exactly zero at the calculated equilibrium potential of −39 mV for 3:1 stoichiometry. This small deviation could be caused by systematic experimental errors, or by leakage or slippage (Parker et al., 2014; Shlosman et al., 2018). To exclude that it was caused specifically by D-aspartate, we repeated the experiment using radiolabeled L-aspartate. The equilibrium potentials for the experiments using D- and L-aspartate were identical, showing that the two stereoisomers use the same coupling stoichiometry.

### Similar mode of enantiomers binding

We determined a crystal structure of Glt$_{Tk}$ in complex with D-aspartate at 2.8 Å resolution (Figure 2A,B). The obtained structure is highly similar to the previously described Glt$_{Tk}$ and Glt$_{Ph}$

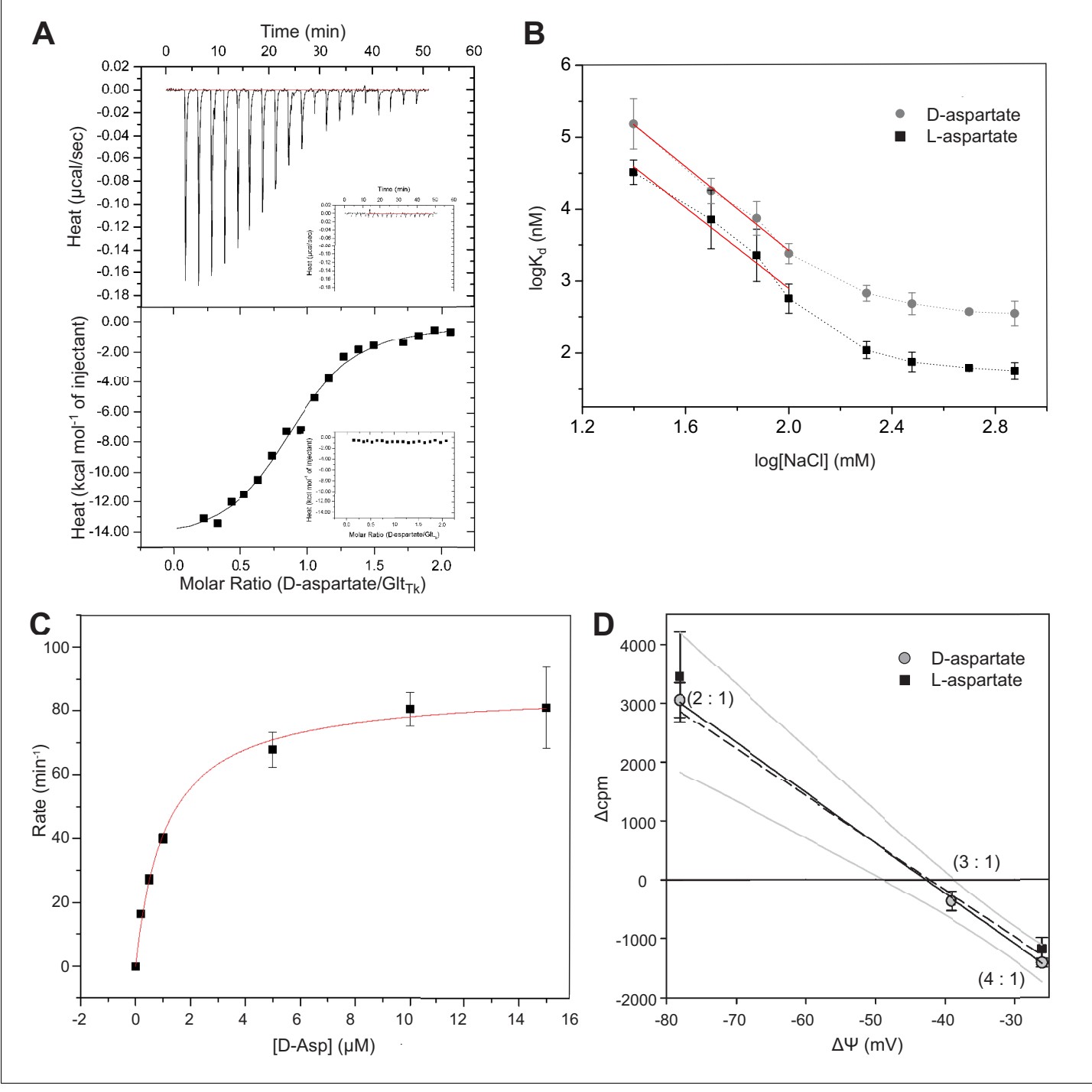

**Figure 1.** Binding and transport of D-aspartate by Glt$_{Tk}$. (**A**) ITC analysis of D-aspartate binding to Glt$_{Tk}$ in presence of 300 mM NaCl ($K_d$ of 0.47 ± 0.17 µM). Insets show no D-aspartate binding in absence of NaCl. (**B**) Sodium and aspartate binding stoichiometry. Logarithmic plot of $K_d$ values (nM) for L-aspartate (black squares; slope is −2.8 ± 0.4; taken for reference from **Guskov et al., 2016**) and D-aspartate (gray circles; slope is −2.9 ± 0.2) against logarithm of NaCl concentration (mM). The negative slope of the double logarithmic plot (red line) in the limit of low sodium concentrations indicates the number of sodium ions that bind together with aspartate. Error bars represent the ±SD from at least three independent measurements. (**C**) Glt$_{Tk}$ transport rate of D-aspartate in presence of 100 mM NaCl. The solid line reports the fit of the Michaelis-Menten model to the data revealing a $K_m$ value of 1.1 ± 0.11 µM. Error bars represent the ±SD from duplicate experiments. (**D**) Determination of Na$^+$ : aspartate coupling stoichiometry in Glt$_{Tk}$ using equilibrium potential measurement. The uptake or efflux of radiolabeled aspartate was determined by comparing the lumenal radioactivity associated with the liposomes after 2 min of incubation with the radioactivity initially present (Δcpm). Gray circles and black squares show the measurements for D- and L-aspartate, respectively. The solid and dashed lines are the best linear regression for the D- and L-aspartate data,

*Figure 1 continued on next page*

*Figure 1 continued*

respectively. The 95% confidence interval for D-aspartate is displayed by gray curves. Numbers in parentheses are the coupling stoichiometries expected to give zero flux conditions for each membrane voltage. Error bars represent the ± SD obtained in five replicates.

DOI: https://doi.org/10.7554/eLife.45286.002

The following source data is available for figure 1:

**Source data 1.** Final concentrations of internal and external buffer used in each reversal potential experiment after diluting the proteoliposomes.

DOI: https://doi.org/10.7554/eLife.45286.003

structures with the transport domains in the outward-oriented occluded state. Comparison of the $Glt_{Tk}$ structures in complex with L- and D-aspartate revealed a highly similar binding mode of the substrates with analogous orientation of amino and carboxyl groups. Despite the impossibility to superimpose two enantiomers, D- and L-aspartate are capable of forming almost identical hydrogen bonding networks with conserved amino acid residues of the substrate-binding site (*Figure 2C*). There are only small changes in the positions of the Cα atoms and Cβ carboxyl groups due to the constitutional differences. However, this divergence leads to only minor changes in the interaction network, consistent with the comparable $K_d$ and ΔH values determined by ITC (*Table 1*).

Three peaks of electron density (*Figure 2D*; *Figure 2—figure supplement 2*) located at the same positions as three sodium ions in the $Glt_{Tk}$ complex with L-aspartate (*Guskov et al., 2016*) most probably correspond to sodium ions, consistent with a 3:1 $Na^+$: D-aspartate coupling stoichiometry (*Figure 1B,D*).

## Discussion

Most proteins selectively bind a single stereoisomer of their substrates (for a review see *Nguyen et al., 2006*). On the other hand, some proteins are able to bind different stereoisomers of a ligand, which is believed to be possible due to different binding modes, because enantiomers cannot be superimposed in the three-dimensional space and thus cannot interact with the binding site identically.

Based on three- and four-point attachment models (*Easson and Stedman, 1933*; *Mesecar and Koshland, 2000*; *Ogston, 1948*) it has been suggested that stereoisomers can bind in the same site but with significant differences. This hypothesis was supported by crystal structures of enzymes with different enantiomeric substrates (*Brem et al., 2016*; *Sabini et al., 2008*), including enantiomeric amino acids (*Aghaiypour et al., 2001*; *Bharath et al., 2012*; *Driggers et al., 2016*; *Temperini et al., 2006*). In contrast, the binding poses of enantiomers in some other enzymes are remarkably similar, for instance in aspartate/glutamate racemase *EcL*-DER, where active site forms pseudo-mirror symmetry (*Liu et al., 2016*).

To our knowledge $Glt_{Tk}$ is the first amino acid transporter for which the binding of enantiomeric substrates has been characterized. The only other transporter for which structures have been determined in the presence of D- and L-substrates is the sodium-alanine symporter AgcS. However, in that case, limited resolution prevented determination of the absolute orientation of bound enantiomers (*Ma et al., 2019*). In the substrate-binding site of $Glt_{Tk}$, L- and D-aspartate take similar poses leading to almost identical networks of contacts. Since mirror imaged substrates inevitably have

**Table 1.** Thermodynamic parameters of D- and L-aspartate binding at high (300 mM) and low (75 mM) $Na^+$ concentration.

| Substrate/ $Na^+$ | $K_d$ (μM) | ΔH (cal mol$^{-1}$) | ΔS (cal mol$^{-1}$ K$^{-1}$) |
|---|---|---|---|
| L-aspartate/300 mM NaCl | 0.12 ± 0.04 | −1.61 (±0.08) x 10$^4$ | −22.1 ± 2.2 |
| D-aspartate/300 mM NaCl | 0.47 ± 0.17 | −1.48 (±0.11) x 10$^4$ | −20.6 ± 3.6 |
| L-aspartate/75 mM NaCl | 1.04 ± 0.39 | −1.22 (±0.13) x 10$^4$ | −13.2 ± 5.2 |
| D-aspartate/75 mM NaCl | 5.66 ± 1.59 | −1.14 (±0.41) x 10$^4$ | −14.3 ± 14.3[*] |

[*]At low $Na^+$ concentrations high errors prevented accurate measuring of ΔS values.

DOI: https://doi.org/10.7554/eLife.45286.004

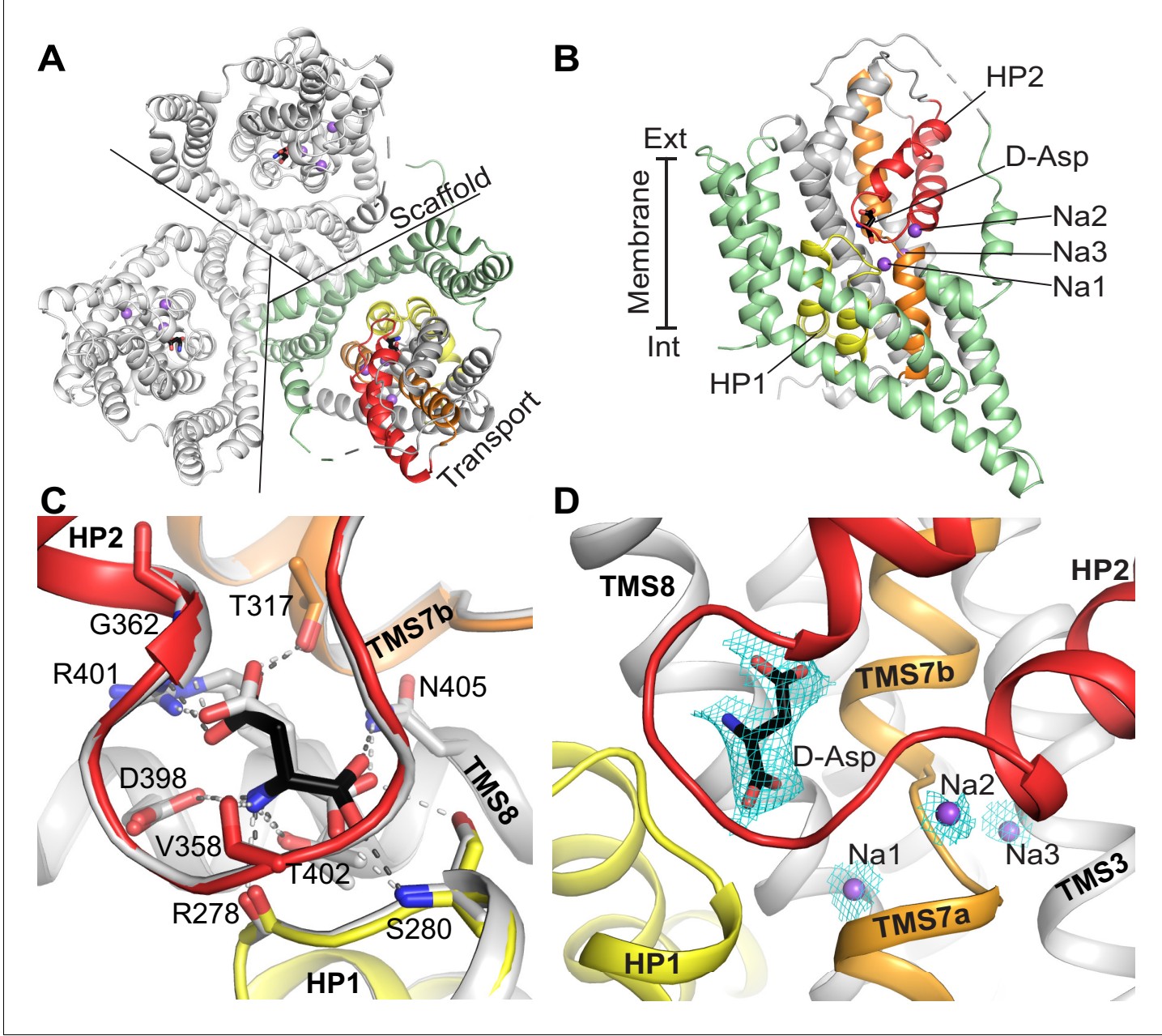

**Figure 2.** The crystal structure of Glt$_{Tk}$ with D-aspartate. The model contains one protein molecule in the asymmetric unit with the substrate present in each protomer of the homotrimer. (**A**) Cartoon representation of the homotrimer viewed from the extracellular side of the membrane. Lines separate protomers. Each protomer consists of the scaffold domain (pale green) and the transport domain. In the transport domain HP1 (yellow), HP2 (red), TMS7 (orange) are shown. D-aspartate is shown as black sticks and Na$^+$ ions as purple spheres. Like in most Glt$_{Ph}$ structures a part of the long flexible loop 3–4 between the transport and scaffold domain is not visible. It is indicated by a dashed connection. (**B**) A single protomer is shown in the membrane plane. (**C**) Comparison of the substrate-binding site of Glt$_{Tk}$ in complex with L-aspartate (gray; PDB code 5E9S) and D-aspartate (black). Cartoon representation; substrates and contacting amino acid residues are shown as sticks; hydrogen bonds are shown as dashed lines. The Glt$_{Tk}$ structures with D- and L-aspartate can be aligned with Cα-RMSD = 0.38 Å for the three transport domains. (**D**) Composite omit map (cyan mesh) for D-aspartate (contoured at 1σ) and sodium ions (2σ) calculated using simulated annealing protocol in Phenix (***Terwilliger et al., 2008***). Color coding in all panels is the same.

DOI: https://doi.org/10.7554/eLife.45286.005

The following figure supplements are available for figure 2:

**Figure supplement 1.** Superposition of substrate-binding sites of L-aspartate bound Glt$_{Tk}$ and thermostabilized human EAAT1.

DOI: https://doi.org/10.7554/eLife.45286.006

*Figure 2 continued on next page*

*Figure 2 continued*

**Figure supplement 2.** Superposition of substrate and sodium binding sites in L-aspartate and D-aspartate bound Glt$_{Tk}$.
DOI: https://doi.org/10.7554/eLife.45286.007
**Figure supplement 3.** Model of glutamate binding in EAAT1.
DOI: https://doi.org/10.7554/eLife.45286.008

differences in angles between donors and acceptors of hydrogen bonds, the binding affinities are not identical, with 4–6 times higher $K_d$ of the Glt$_{Tk}$-D-aspartate complex in comparison with L-aspartate (*Table 1*). Similar differences in binding affinities between these enantiomers were also found for the Glt$_{Ph}$ homologue (*Boudker et al., 2007*). The higher $K_d$ values for the D-aspartate enantiomer might be explained by a higher dissociation rate ($k_{off}$) in comparison with L-aspartate, that was shown in kinetic studies of sodium and aspartate binding on Glt$_{Ph}$ (*Ewers et al., 2013*; *Hänelt et al., 2015*). Glt$_{Tk}$ couples binding and transport of three sodium ions to one D-aspartate molecule (*Figure 1B,D*), the same number as for L-aspartate. Although the affinity for D-aspartate is lower than for L-aspartate, the binding of D-aspartate is not accompanied by a loss of sodium binding sites, which is in line with the observation that none of the sodium binding sites are directly coordinated by the substrate L-aspartate. In the crystal structure of Glt$_{Tk}$ with D-aspartate peaks of density were resolved at positions corresponding to the three sodium ions in the L-aspartate bound Glt$_{Tk}$ structure (*Figure 2D*) (*Guskov et al., 2016*). Altogether our data suggest that the mechanism of D- and L-aspartate transport in Glt$_{Tk}$ is most probably identical.

Mammalian glutamate transporters take up D-aspartate, L-glutamate and L-aspartate with similar micromolar affinity, but have significantly lower affinity (millimolar) for D-glutamate (*Arriza et al., 1997*; *Arriza et al., 1994*). In the absence of the structures of human SLC1A transporters with different stereoisomeric substrates, one can only speculate why EAATs can readily bind and transport both L- and D-aspartate, but only L-glutamate. It seems that the extra methylene group in D-glutamate compared to D-aspartate could cause sterical clashes within the binding site (*Figure 2 — Figure 2—figure supplement 3*), which might affect affinity of binding.

## Materials and methods

**Key resources table**

| Reagent type (species) or resource | Designation | Source or reference | Identifiers | Additional information |
|---|---|---|---|---|
| Gene | TK0986 | UniProt database | Q5JID0 | |
| Strain, strain background (*E. coli*) | MC1061 | *Casadaban and Cohen, 1980* | | |
| Biological sample (*Thermococcus kodakarensis* KOD1) | | | ATCC BAA-918/JCM 12380/KOD1 | |
| Recombinant DNA reagent | pBAD24-Glt$_{Tk}$-His8 | *Jensen et al., 2013* | | Expression plasmid for C-terminally His8-tagged Glt$_{Tk}$. |
| Chemical compound | D-Asp | Sigma-Aldrich | 219096–25G | ReagentPlus99% |
| Software | Origin 8 | OriginLab | | |

*Continued on next page*

Continued

| Reagent type (species) or resource | Designation | Source or reference | Identifiers | Additional information |
|---|---|---|---|---|
| Other | Glt$_{Tk}$-D-aspartate coordinate file and structural factors | This paper | accession number PDB ID code 6R7R | Crystal structure of the glutamate transporter homologue Glt$_{Tk}$ in complex with D-aspartate |

## Protein purification and crystallization

Glt$_{Tk}$ was expressed and purified as described previously (*Guskov et al., 2016*). It was shown that L-aspartate binds to Glt$_{Tk}$ only if sodium ions are present, and the protein purified in absence of sodium ions is in the *apo* state (*Jensen et al., 2013*). For crystallization with D-aspartate the *apo* protein was purified by size exclusion chromatography (SEC) on a Superdex 200 10/300 GL (GE Healthcare) column equilibrated with buffer containing 10 mM Hepes KOH, pH 8.0, 100 mM KCl, 0.15% DM. Crystals of Glt$_{Tk}$ with D-aspartate were obtained in presence of 300 mM NaCl, 300 µM D-aspartate (Sigma-Aldrich, 99%) by the vapour diffusion technique (hanging drop) at 5°C by mixing equal volumes of protein (7 mg ml$^{-1}$) and reservoir solution (20% glycerol, 10% PEG 4000, 100 mM Tris/bicine, pH 8.0, 60 mM CaCl$_2$, 60 mM MgCl$_2$, 0.75% n-octyl-b-D-glucopyranoside (OG)).

## Data collection and structure determination

Crystals were flash-frozen without any additional cryo protection and data sets were collected at 100K at the beamline ID23-1 (ESRF, Grenoble). The data were indexed, integrated and scaled in XDS (*Kabsch, 2010*) and the structure was solved by Molecular Replacement with Phaser (*McCoy et al., 2007*) using structure of Glt$_{Tk}$ (PDB ID 5E9S) as a search model. Manual model rebuilding and refinement were carried out in COOT (*Emsley et al., 2010*) and Phenix refine (*Afonine et al., 2012*). Data collection and refinement statistics are summarized in *Table 2*. Coordinates and structure factors for Glt$_{Tk}$ have been deposited in the Protein Data Bank under accession codes PDB 6R7R. All structural figures were produced with an open-source version of PyMol.

## Isothermal titration calorimetry

ITC experiments were performed at a constant temperature of 25°C using an ITC200 calorimeter (MicroCal). Varying concentrations of the indicated substrates (in 10 mM Hepes KOH, pH 8.0, 100 mM KCl, 0.15% DM and indicated sodium concentrations) were titrated into a thermally equilibrated ITC cell filled with 250 µl of 3–20 µM Glt$_{Tk}$ supplemented with 0 to 1000 mM NaCl. Data were analyzed using the ORIGIN-based software provided by MicroCal.

## Reconstitution into proteoliposomes

A solution of *E. coli* total lipid extract (20 mg ml$^{-1}$ in 50 mM KPi, pH 7.0) was extruded with a 400-nm-diameter polycarbonate filter (Avestin, 11 passages) and diluted with the same buffer to a final concentration of 4 mg ml$^{-1}$. The lipid mixture was destabilized with 10% Triton-X100. Purified Glt$_{Tk}$ and the destabilized lipids were mixed in a ratio of 1:1600 or 1:250 (protein: lipid) and incubated at room temperature for 30 min. Bio-beads were added four times (25 mg ml$^{-1}$, 15 mg ml$^{-1}$, 19 mg ml$^{-1}$, 4 mg ml$^{-1}$ lipid solution) after 0.5 hr, 1 hr, overnight and 2 hr incubation, respectively, on a rocking platform at 4°C. The Bio-beads were removed by passage over an empty Poly-Prep column (Bio-Rad). The proteoliposomes were collected by centrifugation (20 min, 298,906 g, 4°C), subsequently resuspended in 50 mM KPi, pH 7.0 to the concentration of the protein 33.4 µg ml$^{-1}$ and freeze-thawed for four cycles. The proteoliposomes were stored in liquid nitrogen until subsequent experiments.

## Uptake assay

Stored proteoliposomes with reconstitution ratio of 1:1600 were thawed and collected by centrifugation (20 min, 298,906 g, 4°C), the supernatant was discarded and the proteoliposomes were resuspended in buffer containing 10 mM KPi, pH 7.5, 300 mM KCl. The internal buffer was exchanged by three cycles of freezing in liquid nitrogen and thawing, and finally extruded through a polycarbonate

**Table 2.** Data collection and refinement statistics.

| | Glt$_{Tk}$ D-Asp |
|---|---|
| **Data collection** | |
| Space group | P3221 |
| Cell dimensions | |
| a, b, c (Å) | 116.55, 116.55, 314.77 |
| α, β, γ (°) | 90.00, 90.00 120.00 |
| Resolution (Å) | 48.06-2.80 (2.87-2.80)* |
| R$_{meas}$ | 0.11 (>1) |
| CC$_{1/2}$ | 99.9 (11.7) |
| I / σI | 8.40 (0.98) |
| Completeness (%) | 99.3 (98.9) |
| Redundancy | 5 (4) |
| **Refinement** | |
| Resolution (Å) | 2.80 |
| No. reflections | 301,077 |
| R$_{work}$/R$_{free}$ (%)s | 23.4/27.2 |
| No. of atom | |
| Protein | 9262 |
| PEG/detergent | 181/33 |
| Ligand/ion | 27/9 |
| Water | - |
| *B*-factors | |
| Protein | 127 |
| PEG/detergent | 147/174 |
| Ligand/ion | 114/117 |
| Water | - |
| R.m.s. deviations | |
| Bond lengths (Å) | 0.008 |
| Bond angles (°) | 1.162 |

*Values in parentheses are for the highest-resolution shell.
DOI: https://doi.org/10.7554/eLife.45286.009

filter with 400 nm pore size (Avestin, 11 passages). The proteoliposomes were finally pelleted by centrifugation (20 min, 298,906 g, 4°C) and resuspended to the concentration of the protein 625 ng µl⁻¹. 2 µl of proteoliposomes were diluted 100 times in reaction buffer containing 10 mM KPi, pH 7.5, 100 mM NaCl, 200 mM Choline-Cl, 3 µM valinomycin and 0.2–15 µM D-aspartate (each concentration point contained 0.2 µM [³H]-D-aspartate). After 15 s the reaction was quenched by adding 2 ml of ice-cold buffer (10 mM KPi, pH 7.5, 300 mM KCl) and immediately filtered on nitrocellulose filter (Protran BA 85-Whatman filter), finally the filter was washed with 2 ml of quenching buffer. The filters were dissolved in scintillation cocktail and the radioactivity was measured with a PerkinElmer Tri-Carb 2800RT liquid scintillation counter.

## Measuring transporter equilibrium potentials

Stored proteoliposomes with reconstitution ratio of 1:250 were thawed and collected by centrifugation (20 min, 298,906 g, 4°C), the supernatant was discarded and the proteoliposomes were resuspended to a concentration of 10 mg ml⁻¹ of lipids in buffer containing 20 mM Hepes/Tris, pH 7.5, 200 mM NaCl, 50 mM KCl, 10 µM D-aspartate (containing 1 µM [³H]-D-aspartate). The internal buffer was exchanged by freeze-thawing and extrusion as described above. The experiment was

started by diluting the proteoliposomes 20 times into a buffer containing 20 mM Hepes/Tris, pH 7.5, 200 mM NaCl, 3 µM valinomycin, varying concentrations of KCl and Choline Cl were added in order to obtain the desired membrane potential as shown in (*Figure 1—source data 1*).

After 1, 2 and 3 min the reaction was quenched with ice-cold quenching buffer containing 20 mM Hepes/Tris, pH 7.5, 250 mM Choline Cl and immediately filtered on nitrocellulose filter (Protran BA 85-Whatman filter), finally the filter was washed with 2 ml of quenching buffer. The initial amount of radiolabeled aspartate was measured by filtering the proteoliposomes immediately after diluting them in quenching buffer. The filters were dissolved in scintillation cocktail and the radioactivity was measured with a PerkinElmer Tri-Carb 2800RT liquid scintillation counter. The equilibrium, or reversal, potential, $E_{rev}$, for each condition was calculated as described in *Fitzgerald et al. (2017)*.

## Acknowledgements

This work is funded by the Netherlands Organisation for Scientific Research (Vici grant 865.11.001 to DJS and Vidi grant 723.014.002 to AG) and European Research Council starting grant 282083 to DJS. We thank A Garaeva and M Ejby for synchrotron data collection. The European Synchrotron Radiation Facility beamlines ID23-1 and ID29 (Grenoble, France) and EMBL beamlines P13 and P14 (Hamburg, Germany) are acknowledged for beamline facilities. This work has been supported by iNEXT, grant number 653706, funded by the Horizon 2020 programme of the European Commission.

## Additional information

### Funding

| Funder | Grant reference number | Author |
| --- | --- | --- |
| Nederlandse Organisatie voor Wetenschappelijk Onderzoek | 865.11.001 | Dirk J Slotboom |
| European Research Council | 282083 | Dirk J Slotboom |
| Nederlandse Organisatie voor Wetenschappelijk Onderzoek | 723.014.002 | Albert Guskov |

The funders had no role in study design, data collection and interpretation, or the decision to submit the work for publication.

### Author contributions

Valentina Arkhipova, Formal analysis, Supervision, Validation, Investigation, Visualization, Writing—original draft, Project administration, Writing—review and editing; Gianluca Trinco, Formal analysis, Investigation, Methodology, Writing—review and editing; Thijs W Ettema, Formal analysis, Investigation; Sonja Jensen, Resources, Investigation; Dirk J Slotboom, Conceptualization, Resources, Data curation, Formal analysis, Supervision, Funding acquisition, Methodology, Project administration, Writing—review and editing; Albert Guskov, Conceptualization, Data curation, Formal analysis, Supervision, Funding acquisition, Validation, Investigation, Methodology, Writing—review and editing

### Author ORCIDs

Dirk J Slotboom https://orcid.org/0000-0002-5804-9689
Albert Guskov http://orcid.org/0000-0003-2340-2216

### Decision letter and Author response

Decision letter https://doi.org/10.7554/eLife.45286.014
Author response https://doi.org/10.7554/eLife.45286.015

## Additional files

### Supplementary files

• Transparent reporting form
DOI: https://doi.org/10.7554/eLife.45286.010

### Data availability

Diffraction data and the derived model have been deposited in PDB under accession number 6R7R.

The following dataset was generated:

| Author(s) | Year | Dataset title | Dataset URL | Database and Identifier |
|---|---|---|---|---|
| Arkhipova V, Dirk Slotboom | 2019 | Diffraction data and the derived model | https://www.rcsb.org/structure/6R7R | Protein Data Bank, 6R7R |

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
