## [Decision Letter]

Thank you for submitting your article "Binding and transport of D-aspartate by the glutamate transporter homologue Glt_Tk_" for consideration by *eLife*. Your article has been reviewed by two peer reviewers, and the evaluation has been overseen by José D Faraldo-Gómez as Reviewing Editor and Richard Aldrich as the Senior Editor. The following individual involved in review of your submission has agreed to reveal their identity: Simon Newstead (Reviewer #1).

The reviewers have discussed the reviews with one another and the Reviewing Editor has drafted this decision to help you prepare a revised submission. Based on this evaluation, we have decided to invite you submit a revised version of your manuscript, in the form of a Short Report.

Summary:

The manuscript by Arkhipova et al. presents a detailed functional and structural analysis of enantiomeric substrate recognition and transport by Glt_Tk_, an archaeal Na+-driven aspartate transporter and a model system for mammalian EAATs, which are responsible for the clearing of glutamate from the synaptic cleft. The importance of these transporters in regulating neuronal signaling motivates molecular-level studies such as this. Indeed, D-aspartate is a potential modulator of neuronal signaling; Arkhipova et al. seek to elucidate how these kind of transporters recognize and translocate both the L- and D- forms of the amino acid. EAATs, as well as Glt_Ph_ (another archaeal member of the same family) have been shown previously to transport both L- and D-aspartate with comparable affinity, suggesting a similar mechanism of recognition and/or transport for the two enantiomers. To gain further insights into these processes, the authors determine the crystal structure of Glt_Tk_ in complex with D-aspartate at 2.8 A resolution, and compare its binding-site configuration to that of the L-aspartate-bound transporter. They find, indeed, that the two structures are virtually identical; the substrates adopt a highly similar binding mode, and the three peaks of electron density tentatively assigned to Na^+^ ions coincide with the positions of Na^+^ binding sites in the L-aspartate structure. This result is further supported by functional experiments that quantify thermodynamic and kinetic parameters of D-aspartate transport. D-aspartate is found to bind Glt_Tk_ with comparable affinity, and to be transported with the same rate and 3:1 Na+:D-Asp stoichiometry as L-aspartate, confirming similar transport mechanism for the two enantiomers.

All reviewers agree that the manuscript presents a careful, self-consistent and highly quantitative investigation. The experimental work is thorough and of high quality, and the underlying research question and the authors' findings are of interest to multiple segments of the readership of *eLife*.

Essential revisions:

1) The reviewers agree that manuscript is at times insufficiently detailed or imprecise, both in the presentation of the data, its explanation and conclusions:

The Kd values for L- and D-aspartate are substantially different, 62 vs. 380 nM (the standard deviation for these measurements should be shown). The authors argue that the "higher Kd values for the D-aspartate enantiomer might be explained by a higher dissociation rate (k_off_) in comparison with L-aspartate, that was shown in kinetic studies of sodium and aspartate binding on Glt_Ph_ (Ewers et al., 2013; Hänelt et al., 2015)." Can the authors not undertake a similar analysis and show whether a difference in the k_on_ or K_off_ does explain the difference, as proposed? Given that explaining the mechanism for enantiomeric selection/recognition (or lack thereof) is the key aim of this study, the manuscript would be strengthen if the reason for this difference was established more conclusively.

The rationale and interpretation of the experiments reported in Figure 3 ought to be described and discussed in greater detail, if necessary using supplementary materials. For example, the mathematical framework underlying the design of the experiments and the interpretation of the data would be informative for non-specialists. In addition, how do the authors interpret that the zero-flux condition is not observed at the predicted membrane voltage values, but at -48 mV? In their view, does this result reflect a variable transport stoichiometry or the occurrence of uncoupled transport? Or does it owe to experimental error/uncertainty? If the latter is more plausible, what are the possible sources of error? Recent studies that include similar assays (for example J Gen Physiol. 2018 Jan 2; 150(1): 51-65) may be used as a reference for additional analysis or discussion.

2) Notwithstanding the quality of the experimental work, as it stands the study appears to be largely confirmative. It would be important that the authors illustrate better the broader physiological or mechanistic significance of their findings.

For example, the authors seem to propose that their results shed light on the general recognition mechanism of L- and D-aspartate by their respective targets, and on how promiscuity and/or selectivity for these stereoisomers are established on the structural level. The authors might consider discussing in greater detail how the binding site configuration of Glt_Tk_ compares to/is distinct from those of both highly selective and non-selective enzymes (e.g. L-asparaginase, mentioned in the Discussion).

Alternatively, the authors could elaborate on how their study fits in the context of substrate recognition and physiology of mammalian EAAT transporters. They draw attention to the fact that their result explains the "surprising" finding that the EAATs recognize and transport both L- and D-aspartate. What might be an even 'curiouser' observation is that the EAATs are unable to distinguish between L- and D-aspartate, yet transport L-glutamate preferentially over D-glutamate. It would be of interest to compare how transporters of this family recognize aspartate vs. glutamate, and establish what structural features allow enantiomeric promiscuity for aspartate, but not for glutamate, a substrate that is different only by one -CH2 group.

---

## [Author Response]

Essential revisions:1) The reviewers agree that manuscript is at times insufficiently detailed or imprecise, both in the presentation of the data, its explanation and conclusions:The Kd values for L- and D-aspartate are substantially different, 62 vs. 380 nM (the standard deviation for these measurements should be shown).

The standard deviations of Kd values are added (subsection “Affinity of D-aspartate and stoichiometry of sodium binding to Glt_Tk_”, first paragraph).

The authors argue that the "higher Kd values for the D-aspartate enantiomer might be explained by a higher dissociation rate (k_off_) in comparison with L-aspartate, that was shown in kinetic studies of sodium and aspartate binding on Glt_Ph_ (Ewers et al., 2013; Hänelt et al., 2015)." Can the authors not undertake a similar analysis and show whether a difference in the k_on_ or K_off_ does explain the difference, as proposed? Given that explaining the mechanism for enantiomeric selection/recognition (or lack thereof) is the key aim of this study, the manuscript would be strengthen if the reason for this difference was established more conclusively.

We performed similar experiments with Glt_Tk_ using tryptophan fluorescence measurements. Unfortunately, whereas fluorescence of the Glt_Ph_ mutant (GltPh F273W) was quenched by binding of L-aspartate to the Na^+^-bound protein (Hänelt et al., 2015), the corresponding Glt_Tk_ mutant F275W/W428F did not show a change in fluorescence upon addition of L-aspartate (Author response image 1). Apparently, minor differences in the environment of the engineered tryptophans in the two proteins lead to differences in the fluorescence quenching properties. Since the quenching is essential for the stopped-flow kinetic measurements, we could not establish that the k_on_ and k_off_ values are similarly affected in Glt_Tk_ and Glt_Ph_. Nonetheless, the overall similarity in structure and transport mechanism of Glt_Tk_ and Glt_Ph_ justifies the speculation that a difference in k_off_ between L- and D-aspartate dissociation is the most likely cause of the differences in Kd values.

**Author response image 1. respfig1:** No change in tryptophan fluorescence of Glt_Tk_ double mutant F275W/W428F observed upon addition of L-aspartate. Fluorescence emission spectra of the protein after the addition of addition of 1M NaCl (black squares) followed by 100 µM L-aspartate (red circles).

The rationale and interpretation of the experiments reported in Figure 3 ought to be described and discussed in greater detail, if necessary using supplementary materials. For example, the mathematical framework underlying the design of the experiments and the interpretation of the data would be informative for non-specialists. In addition, how do the authors interpret that the zero-flux condition is not observed at the predicted membrane voltage values, but at -48 mV? In their view, does this result reflect a variable transport stoichiometry or the occurrence of uncoupled transport? Or does it owe to experimental error/uncertainty? If the latter is more plausible, what are the possible sources of error? Recent studies that include similar assays (for example J Gen Physiol. 2018 Jan 2; 150(1): 51-65) may be used as a reference for additional analysis or discussion.

We have extended the section on rationale and interpretation of the reversal potential measurements in the last paragraph of the subsection “Affinity of D-aspartate and stoichiometry of sodium binding to Glt_Tk_”. We also mention possible causes of the slight deviation of the reversal potential from the predicted value for 3:1 stoichiometry, and include references. In addition, we have included 95% confidence intervals for the regression shown in new Figure 1D. This analysis shows that the predicted value of the reversal potential for 3:1 stoichiometry falls within the interval. Nonetheless, we do not know why the zero-flux condition is not observed exactly at the predicted membrane voltage. However, for our conclusions, the more important question is whether there is a difference between L- and D-aspartate. Therefore, we have performed the same reversal potential measurements with L-aspartate as substrate (included in new Figure 1D). The reversal potentials for measurements using L- and D- aspartate are identical.

2) Notwithstanding the quality of the experimental work, as it stands the study appears to be largely confirmative. It would be important that the authors illustrate better the broader physiological or mechanistic significance of their findings.For example, the authors seem to propose that their results shed light on the general recognition mechanism of L- and D-aspartate by their respective targets, and on how promiscuity and/or selectivity for these stereoisomers are established on the structural level. The authors might consider discussing in greater detail how the binding site configuration of Glt_Tk_ compares to/is distinct from those of both highly selective and non-selective enzymes (e.g. L-asparaginase, mentioned in the Discussion).

In the light of the shortening of the manuscript to a Short Report, as requested by the editor, we have rewritten, the paragraph on the comparison for clarity, but have not extended the Discussion.

Alternatively, the authors could elaborate on how their study fits in the context of substrate recognition and physiology of mammalian EAAT transporters. They draw attention to the fact that their result explains the "surprising" finding that the EAATs recognize and transport both L- and D-aspartate. What might be an even 'curiouser' observation is that the EAATs are unable to distinguish between L- and D-aspartate, yet transport L-glutamate preferentially over D-glutamate. It would be of interest to compare how transporters of this family recognize aspartate vs. glutamate, and establish what structural features allow enantiomeric promiscuity for aspartate, but not for glutamate, a substrate that is different only by one -CH2 group.

We decided to discuss in greater details EAAT transporters (rather than L- and D-selective enzymes), because it is more relevant for our conclusions (Introduction, third paragraph and Discussion, last paragraph). We emphasize that this discussion is still highly speculative and gives only hints of an explanation (steric hindrance) for the ‘curiouser’ observation. We have added Figure 2—figure supplement 1, where we show a comparison of the EAAT1 and Glt_Tk_ substrate-binding sites, and Figure 2—figure supplement 3 where we modeled both L- and D- glutamate in the EAAT1 substrate binding site.